# The value of artificial intelligence in the diagnosis of lung cancer: A systematic review and meta-analysis

Mingsi Liu[1]☯, Jinghui Wu[2]☯, Nian Wang[3], Xianqin Zhang[3], Yujiao Bai[3,4], Jinlin Guo[5], Lin Zhang ORCID[6]*, Shulin Liu[7]*, Ke Tao ORCID[2]*

1 Department of Computer and Artificial Intelligence, Zhengzhou University, Zhengzhou, Henan, China, 2 College of Life Science, Sichuan University, Chengdu, Sichuan, China, 3 School of Basic Medical Sciences, Chengdu Medical College, Chengdu, Sichuan, China, 4 Non-Coding RNA and Drug Discovery Key Laboratory of Sichuan Province, Chengdu Medical College, Chengdu, Sichuan, China, 5 Chongqing Key Laboratory of Sichuan-Chongqing Co-construction for Diagnosis and Treatment of Infectious Diseases Integrated Traditional Chinese and Western Medicine, Chengdu University of Traditional Chinese Medicine, Chengdu, China, 6 Department of Pharmacy, Shaoxing people's Hospital, Shaoxing, Zhejiang, China, 7 Department of the First Affiliated Hospital of Chengdu Medical College, Sichuan, China

☯ These authors contributed equally to this work.
* taoke@scu.edu.cn (KT); zhanglinfudan@zju.edu.cn (LZ); 928875999@qq.com (SL)

**Data Availability Statement:** All relevant data are within the paper and its Supporting Information files.

## Abstract

Lung cancer is a common malignant tumor disease with high clinical disability and death rates. Currently, lung cancer diagnosis mainly relies on manual pathology section analysis, but the low efficiency and subjective nature of manual film reading can lead to certain misdiagnoses and omissions. With the continuous development of science and technology, artificial intelligence (AI) has been gradually applied to imaging diagnosis. Although there are reports on AI-assisted lung cancer diagnosis, there are still problems such as small sample size and untimely data updates. Therefore, in this study, a large amount of recent data was included, and meta-analysis was used to evaluate the value of AI for lung cancer diagnosis. With the help of STATA16.0, the value of AI-assisted lung cancer diagnosis was assessed by specificity, sensitivity, negative likelihood ratio, positive likelihood ratio, diagnostic ratio, and plotting the working characteristic curves of subjects. Meta-regression and subgroup analysis were used to investigate the value of AI-assisted lung cancer diagnosis. The results of the meta-analysis showed that the combined sensitivity of the AI-aided diagnosis system for lung cancer diagnosis was 0.87 [95% CI (0.82, 0.90)], specificity was 0.87 [95% CI (0.82, 0.91)] (CI stands for confidence interval.), the missed diagnosis rate was 13%, the misdiagnosis rate was 13%, the positive likelihood ratio was 6.5 [95% CI (4.6, 9.3)], the negative likelihood ratio was 0.15 [95% CI (0.11, 0.21)], a diagnostic ratio of 43 [95% CI (24, 76)] and a sum of area under the combined subject operating characteristic (SROC) curve of 0.93 [95% CI (0.91, 0.95)]. Based on the results, the AI-assisted diagnostic system for CT (Computerized Tomography), imaging has considerable diagnostic accuracy for lung cancer diagnosis, which is of significant value for lung cancer diagnosis and has greater feasibility of realizing the extension application in the field of clinical diagnosis.

**Funding:** This study was supported by two national natural science foundation of China projects (grant no. 32170119,31870135). The funders had no role in study design, data collection and analysis, decision to publish, or preparation of the manuscript.

# 1 Introduction

Lung cancer is a malignant tumor originates from the epithelial tissue. Air pollution, radiation exposure, and fungal infections contribute to the persistent incidence and mortality of lung cancer, which ranks first in incidence and mortality among malignant tumors in China and worldwide. The overall five-year survival rate of lung cancer is only 15.6%, and the prognosis of patients with different clinical stages is significantly different. The cure rate of carcinoma in situ is close to 100%, however, according to statistics from relevant sources, the 5-year survival rate of lung cancer is 70% in stage I, and less than 5% in stage IV, respectively [1]. Therefore, early diagnosis of lung cancer is critical. In Table 1, we provide the specific meanings of the

**Table 1. Symbol interpretation table.**

| Symbols | They stands for |
|---|---|
| CI | confidence interval |
| CT | Computerized Tomography |
| MRI | Magnetic Resonance Imaging |
| PET-CT | Positron Emission Tomography-Computed Tomography |
| SSAC | Semi-Supervised Adversarial Classification |
| MV-KBC | Multi-View Knowledge-Based Collaborative |
| LDSCT | Low-Dose Spiral Computed Tomography |
| WLB | white light bronchoscopy |
| AFB | fluorescence bronchoscopy |
| FCFM | fluorescence confocal microscopy |
| EBUS | endobronchial ultrasound |
| VNB | virtual navigational bronchoscopy |
| ENB | electromagnetic navigational bronchoscopy |
| pro-GRP | gastrin-releasing peptide precursor |
| NSE | neuron-specific enolase |
| CEA | carcinoembryonic antige |
| VOCs | volatile organic compounds |
| NSCLC | non-small cell lung cancer |
| GANs | generative adversarial networks |
| DCNNs | deep convolutional neural networks |
| AUC | Area Under The Curve |
| lncRNAs | long non-coding RNAs |
| SROC | combined subject operating characteristic curve |
| PRISMA | Preferred Reporting Items for Systematic Reviews and Meta-Analyses |
| STARD | The Standards for Reporting of Diagnostic Accuracy |
| CBM | China Biomedical Literature Database |
| CNKI | China Knowledge Network |
| FP | number of false positives |
| Sen | sensitivity |
| TP | number of true positives |
| Spe | specificity |
| FN | number of false negatives |
| Acc | accuracy |
| TN | number of true negatives |
| +LR combined | positive likelihood ratio |
| -LR combined | negative likelihood ratio |
| DOR combined | diagnostic ratio |

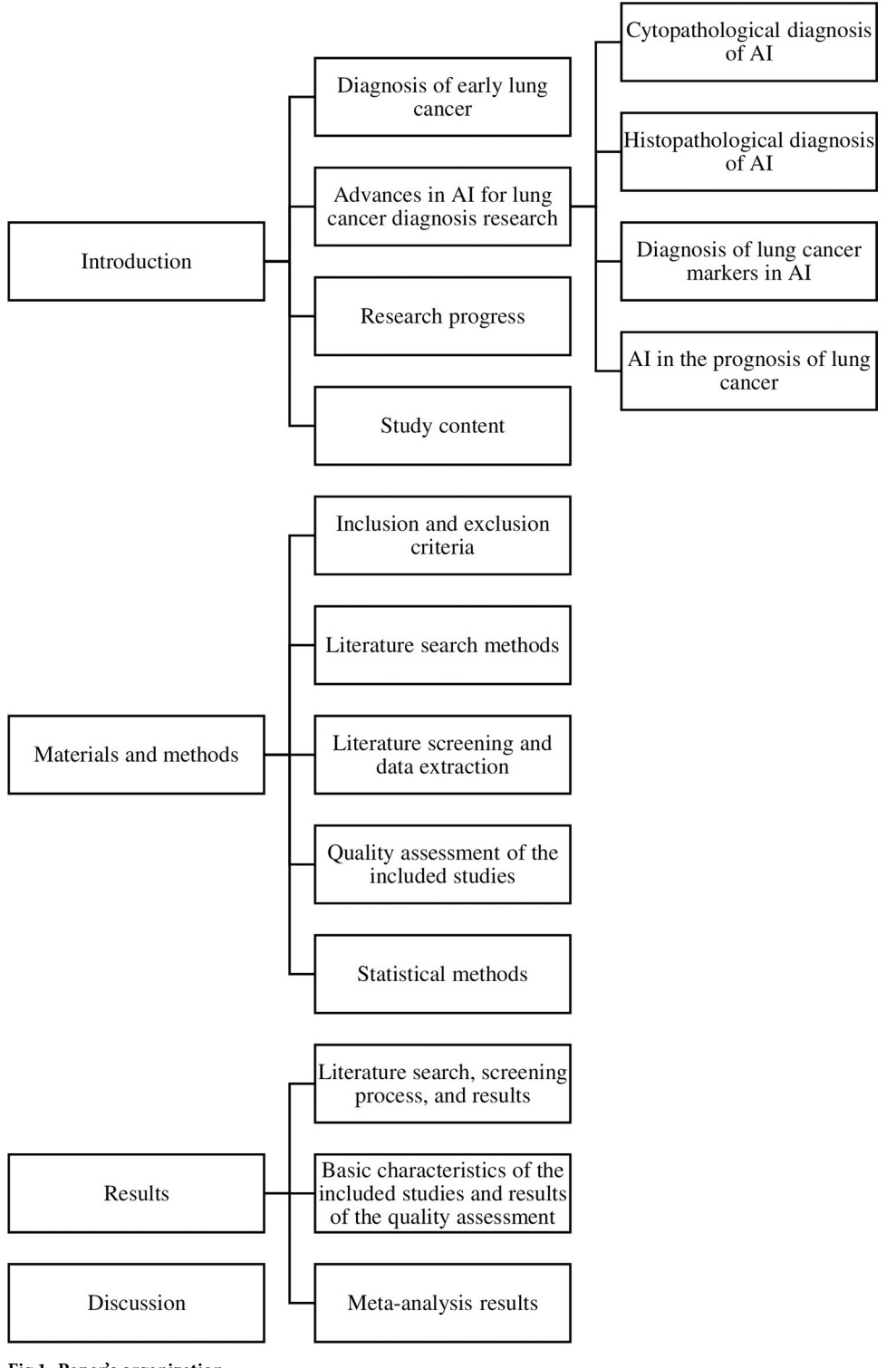

**Fig 1. Paper's organization.**

abbreviations used throughout the paper. In Fig 1, we present the overall structure of the paper.

## 1.1 Diagnosis of early lung cancer

In recent years, techniques and methods for early lung cancer diagnosis have been broadly divided into three main categories: diagnostic imaging, diagnostic pathology tests, and diagnostic marker tests. Diagnostic imaging refers to the use of chest radiographs, chest CT, chest MRI (Magnetic Resonance Imaging), and PET-CT (Positron Emission Tomography-Computed Tomography) in the chest to identify the presence of a mass or lesion in the lung and determine the possibility of cancer [2]. In the 1990s, Naildich [3] proposed LDSCT (Low-Dose Spiral Computed Tomography) as a new method of lung cancer screening. LDSCT is significantly more sensitive than chest radiography for early lung cancer. Related studies in the US, Japan, and Germany have indicated that LDSCT can reduce patients' radiation dose and have a higher sensitivity for lung nodule detection [4]. However, the imaging diagnosis is easily affected by factors such as the patient's lung vacuolation and lung tissue reshaping, and the diagnostic accuracy is relatively low, so its application in early-stage lung cancer is minimal [5].

Pathological diagnosis often refers to histological examination through bronchoscopy or percutaneous puncture biopsy, which is the gold standard for lung cancer diagnosis. The main methods of obtaining histological specimens are fibreoptic bronchoscopy, ultrasound, or CT-guided percutaneous lung biopsy, among which bronchoscopy is the most commonly used and growing fastest in recent years [6]. Fibreoptic bronchoscopy is the essential means to confirm the diagnosis of lung cancer. However, the traditional white light bronchoscopy (WLB) has a meager diagnostic rate (<29%) for peripheral lung cancer, especially for some early mucosal and submucosal lesions and precancerous lesions [7]. Therefore, the rise of new bronchoscopic techniques in recent years, such as fluorescence bronchoscopy (AFB), fluorescence confocal microscopy (FCFM), endobronchial ultrasound (EBUS), virtual navigational bronchoscopy (VNB), and electromagnetic navigational bronchoscopy (ENB), have expanded the diagnostic field and improved the diagnostic rate, especially playing a vital role in the diagnosis of early-stage lung cancer. However, most of these techniques have limitations. For example, they are susceptible to external factors, have a high false-positive rate, or still need further experiments to verify whether they can be used in clinical practice.

Lung cancer marker testing refers to tumor markers such as glycoprotein substances, enzymes, and hormonal substances expressed and secreted by tumor tissue that can be obtained through blood and body fluid tests [8]. At present, dozens of tumor markers associated with lung cancer have been identified, such as gastrin-releasing peptide precursor (pro-GRP), neuron-specific enolase (NSE), and carcinoembryonic antigen (CEA), squamous cell carcinoma antigen (SCC-Ag), etc. [9]. However, a tumor marker with high sensitivity and specificity has not yet been applied to the clinical independently. Therefore, more research has been conducted to detect multiple tumor markers from different tissue sources such as pleural fluid, serum, and bronchial lavage fluid simultaneously, which is expected to improve the diagnostic rate of tumor markers for early lung cancer.

In addition, many novel markers can be used for the early detection and diagnosis of lung cancer, such as molecular biomarkers of susceptibility-related genes oncogenes LOH, DNA, methylated telomerase, and exhaled volatile organic compounds (VOCs) in high-risk groups. However, there are many different types of VOCs, complex sources, and many elements of detection technology involved, and there is still a lack of uniform standards [10]. Once standardized detection techniques are identified, the method will revolutionize the early screening of lung cancer.

## 1.2 Advances in AI for lung cancer diagnosis research

The current clinical practice of early screening using CT scans of the chest is a time-consuming and relatively subjective process that is prone to inter-observer variability. With the

development of AI and digital pathology in recent years, the medical community has increasingly recognized the significant clinical and scientific value of AI in aiding pathological diagnosis. The application of AI recognition technology enables multi-parametric clustering analysis to help physicians screen for early-stage lung cancer [11], reducing errors and increasing problem-solving efficiency. AI has made breakthroughs in detecting, diagnosing, and treating lung cancer [12].

**1.2.1 Cytopathological diagnosis of AI.** Lung cancer is divided into small cell lung cancer, non-small cell lung cancer (NSCLC), and NSCLC accounts for most lung cancers and is the primary pathological type of lung cancer death [13]. AI cytopathology diagnostic systems have been applied to lung cancer classification and diagnosis. Teramoto [14] developed a method to automatically generate cytological images using generative adversarial networks (GANs) to improve deep convolutional neural networks (DCNNs) by using actual and synthetic cytological images and GANs. Neural networks (DCNNs) use authentic and synthetic cytological images and generative adversarial networks to improve the classification accuracy of DCNNs. The results show a substantial increase in accuracy compared to previous studies that did not use GAN-generated images for pre-training. These results confirm the effectiveness of their proposed method for the classification of cytological images when only limited data is available.

**1.2.2 Histopathological diagnosis of AI.** In lung cancer histopathological diagnosis, AI has been able to accurately classify lung cancer subtypes through analysis of digital pathological tissue sections and can predict the survival prognosis of NSCLC patients. Yu [15] used images of tissue sections from lung adenocarcinoma and squamous cell carcinoma patients for validation, extracted morphological image features and developed classifiers that effectively distinguish malignant tumors from adjacent healthy tissues (AUC = 0.81). In addition, it was able to accurately predict long-term survival in patients with stage I adenocarcinoma (log-rank test P = 0.002) and squamous cell carcinoma (log-rank test P = 0.023). Coudray [16] trained a deep convolutional neural network on the full range of section images obtained from The Cancer Genome Atlas, which accurately and automatically classified lung histopathological images into adenocarcinoma, squamous cell carcinoma, and normal lung tissue with results consistent with the pathologist's analysis, with an average AUC (Area Under The Curve) of 0.97.

**1.2.3 Diagnosis of lung cancer markers in AI.** In addition to detecting malignant lung lesions using imaging histology, tumor markers have a crucial role in cancer detection. Specific long non-coding RNAs (lncRNAs) have promoted or inhibited cancer progression in lung cancer patients and are expected to be used as diagnostic markers. Wang [17] used machine learning and weighted gene co-expression networks, the Lasso algorithm, and other techniques to screen the lung genome atlas database of 1364 lncRNAs for the adenocarcinoma best biological markers. LANCL1-AS1, MIR3945HG, and LINC01270 were identified as LUAD markers, with MIR3945HG also the biomarker with the highest diagnostic value for LUSC and strongly correlated with survival [18].

Detection of mutated genes is also now a routine and essential part of the treatment and prognosis of lung cancer. It has been demonstrated that AI can help detect mutated genes in lung cancer. Coudray [19] hypothesized that specific gene mutations would alter the arrangement of lung cancer tumor cells on whole-section images. Thus they predicted the ten most common mutated genes in adenocarcinoma by training a neural network and found that six (STK11, EGFR, FAT1, SETBP1, KRAS, and TP53) could be predicted by pathological images with an accuracy of 73.3% to 85.6%. The study illustrates that the AI histopathology diagnostic system has the potential to help pathologists rapidly detect mutated genes in lung cancer, facilitating the early initiation of targeted drug therapy to improve treatment outcomes and patient prognosis.

**1.2.4 AI in the prognosis of lung cancer.** Wang [20] developed a ConvPath automated cell classification model, which can output microenvironmental characteristics of tumors based on the spatial distribution characteristics of different cell types, and the microenvironmental characteristic was shown to be an independent prognostic factor for lung cancer adenocarcinoma. Their studies also demonstrated that patients' survival could be predicted by analyzing the spatial organization of 48 HE-stained images of lung adenocarcinoma. Late survival rates were significantly lower in the high-risk group than in the low-risk group. These studies illustrate that AI diagnostic systems can yield valuable quantitative information for the prognosis of lung cancer patients.

## 1.3 Research progress

In recent years, with satisfactory results, AI lung cancer screening has been used in the lung cancer population. Huang [21] investigated a novel diagnostic method based on deep transfer convolutional neural networks and extreme learning machines to deal with benign and malignant nodule classification, and the diagnostic method had an accuracy of 94.57% and an AUC of 0.95. In this study, the AI-assisted diagnostic system diagnosed lung cancer with a combined sensitivity of 0.86, the specificity was 0.88, the positive likelihood ratio was 7.2, the negative likelihood ratio was 0.16, the diagnostic ratio was 46, and the AUC value was 0.93. Irvin [22] used a deep learning algorithm called CheXNeXt to improve diagnostic accuracy for lung cancer on chest radiographs. The results showed that the sensitivity of the AI detection method for diagnosis in lung cancer populations was 0.899, the specificity was 0.901, and the AUC value was 0.935. It indicates that AI lung cancer screening for lung cancer populations can obtain high and that using AI algorithms to assist in the diagnosis of lung cancer can help to reduce the time required for radiologists to review images. It can provide an imaging basis and reference for clinical diagnosis and treatment, which is consistent with the results of this study. Dong [23] conducted a value analysis of the AI-assisted diagnostic system based on CT images for 4771 lung cancer diagnoses. The results showed that the combined sensitivity, specificity, positive likelihood ratio, negative likelihood ratio, diagnostic ratio, and AUC values were 0.87, 0.89, 7.70, 0.14, 53.54, and 0.94. The findings were similar to the present study, indicating that the AI-assisted diagnostic system has a high diagnostic value. The results showed that the AI-assisted diagnostic system has a high diagnostic value for lung cancer and can be used to diagnose lung cancer in the clinical setting. Xie [24] et al. designed Semi-Supervised Adversarial Classification (SSAC) model that can be trained with limited labeled and unlabeled data. The model used Multi-View Knowledge-Based Collaborative (MV-KBC), and achieved 92.53% accuracy and 96.28% specificity in LIDC-IDRI database. This suggests that the results of AI lung cancer screening are not limited by the physician's expertise, which is consistent with the findings of this study, and that AI-assisted lung cancer screening improves the sensitivity and accuracy of early lung cancer identification, aids clinicians in diagnosis, and reduces physician workload.

## 1.4 Study content

So far, there have been many studies on the effectiveness of AI-assisted diagnostic systems based on CT images in the diagnosis of lung cancer, but these studies have a small sample size, different study quality, and different AI algorithms. Therefore, this paper adopts the meta-analysis method to systematically evaluate and meta-analyze the diagnostic value of AI-assisted diagnostic systems in lung cancer, in order to provide evidence for clinical application. This paper expands the sample size and uses meta-analysis to evaluate the value of the AI diagnostic system for lung cancer diagnosis. With the help of STATA16.0, the importance of AI-aided

diagnosis for lung cancer diagnosis was assessed by combining effect measures, including specificity, sensitivity, negative likelihood ratio, positive likelihood ratio, diagnostic ratio, and plotting the operational characteristics curve of subjects, and meta-regression. The study also explored the reasons for the heterogeneity between studies employing meta-regression and subgroup analysis to provide evidence for clinical application.

The results of the meta-analysis showed that the combined sensitivity of the AI-aided diagnosis system for lung cancer diagnosis was 0.87 [95% CI (0.82, 0.90)], specificity was 0.87 [95% CI (0.82, 0.91)] (CI stands for confidence interval.), the missed diagnosis rate was 13%, the misdiagnosis rate was 12%, the positive likelihood ratio was 6.5 [95% CI (4.6, 9.3)], the negative likelihood ratio was 0.15 [95% CI (0.11, 0.21)], a diagnostic ratio of 43 [95% CI (24, 76)] and a sum of area under the combined subject operating characteristic (SROC) curve of 0.93 [95% CI (0.91, 0.95)]. Therefore, this study shows that in clinical practice, AI recognition technology can effectively improve the diagnostic sensitivity of early lung cancer, assist physicians to screen early lung cancer more effectively and quickly, and become an auxiliary tool for clinical diagnosis of lung cancer, worthy of clinical promotion.

## 2 Materials and methods

This meta-analysis was based on the Cochrane Handbook for Systematic Reviews (v5.1.0) and Preferred Reporting Items for Systematic Reviews and Meta-Analyses (PRISMA) NMA Checklist [25].

### 2.1 Inclusion and exclusion criteria

Inclusion criteria: (1) Screening articles were following the STARD (The Standards for Reporting of Diagnostic Accuracy) statement; (2) Published literature containing AI reading of chest CT images were used to diagnose lung cancer; (3) AI was used to read chest CT images with precise diagnostic results; (4) The specificity and sensitivity of AI reading chest CT images for the diagnosis of lung cancer is based on pathology testing as the gold standard.

Exclusion criteria: (1) literature for which complete data and duplicates are not available; (2) literature for which pathology testing is not the gold standard; (3) literature for which case studies, reviews, animal studies, reviews, abstracts, etc. are available.

### 2.2 Literature search methods

A comprehensive search of Chinese and English databases was conducted to retrieve literature on AI reading of chest CT images for the diagnosis of lung cancer from January 1, 2010 to August 2021. Chinese databases include: China Biomedical Literature Database (CBM), Wanfang Database, and China Knowledge Network (CNKI); English databases include: PubMed, EMbase, the Cochrane Library, This study combined a search for subject terms, keywords, or free words.

Our search strategies were as follows: "Pulmonary Neoplasms* OR Neoplasms, Lung* OR Lung Neoplasm* OR Neoplasm, Lung* OR Neoplasms, Pulmonary* OR Neoplasm, Pulmonary* OR Pulmonary Neoplasm* OR Lung Cancer* OR Cancer, Lung* OR Cancers, Lung* OR Lung Cancers* OR Pulmonary Cancer* OR Cancer, Pulmonary;Cancers, Pulmonary* OR Pulmonary Cancers* OR Cancer of the Lung* OR Cancer of Lung" AND "Intelligence, Artificial OR Computational Intelligence OR Intelligence, Computational OR Machine Intelligence OR Intelligence, Machine OR Computer Reasoning OR Reasoning, Computer OR AI (Artificial Intelligence) OR Computer Vision Systems OR Computer Vision System OR System, Computer Vision OR Systems, Computer Vision OR Vision System, Computer OR Vision Systems, Computer OR Knowledge Acquisition (Computer) OR Acquisition,

Knowledge (Computer) OR Knowledge Representation (Computer) OR Knowledge Representations (Computer) OR Representation, Knowledge (Computer) OR mechanical ventilation" AND "randomized controlled trial* OR RCT*". The search strategy are provided in Appendix 1–6.

## 2.3 Literature screening and data extraction

Storing and removing duplicate literature was done by EndNote X7 software. Two researchers were selected to screen the literature independently, and where there was disagreement, the agreement was reached after a discussion between the two researchers. Data collection included: authors, date of publication, number of cases, general patient information, AI algorithms, diagnostic criteria, classification models, labeling methods, processed images, features, database sources, etc., number of false positives (FP), sensitivity (Sen), number of true positives (TP), specificity (Spe), number of false negatives (FN), accuracy (Acc), number of true negatives (TN), etc.

## 2.4 Quality assessment of the included studies

The quality of the studies included in this study was assessed by reference to the QUADAS-2 tool, with the software Revman Manager(V5.3). The risk of bias was set in order according to the four components described on the website when analysing with reference to the QUADAS-2 tool, and this phase consisted of two main tasks, firstly comparing the information in the included literature against the QUADAS-2 device and answering each question with the available answers of "yes" After this has been completed, the second part of the work provides a ranking of the risk of bias according to the QUADAS-2 tool, together with information from the included literature as "high risk", "U", "U" and "U". High risk", "Unclear risk" or "Low risk", judged according to the following criteria: if the answer to all questions in a section is " If the answers to the questions in a section are all "yes", then "Low risk" can be selected, if there is a "no" answer in a section, then the risk of bias is considered high and "High risk "If "unclear" appears, the first two principles are followed, and if no result is found, the classification is based on "unclear risk". Finally, feedback on the risk of bias and quality score is provided in a quality assessment chart.

## 2.5 Statistical methods

Statistical analysis was carried out using Stata 16.0 software. The $I^2$ value test was used to determine the high or low heterogeneity to select an appropriate effect model. If $I^2 < 50\%$, the heterogeneity among the included studies was considered inadequate, and a fixed-effects model was used for merging; if $I^2 \geq 50\%$, the heterogeneity among the included studies was deemed high a random-effects model should be used for connecting. A $2*2$ four-compartment table of the AI-assisted diagnostic system for lung cancer diagnosis was presented for each included literature in turn, according to the gold standard of each included literature. The data from all the included literature were integrated to obtain the four-grid tables. The combined effect sizes were calculated, including combined sensitivity (Sen combined), combined specificity (Spe combined), positive likelihood ratio (+LR combined), negative likelihood ratio (-LR combined), diagnostic ratio (DOR combined), and 95% confidence interval (95% CI) of the above data, and the AUC was calculated by plotting the SROC curve. The diagnostic value of the AI-assisted diagnostic system for lung cancer was quantitatively evaluated using the above data. It was considered to have no diagnostic value when the AUC value was in the [0,0.5] range, a low diagnostic value when it was in the [0.5,0.7] field, a high diagnostic value when it was in the [0.7,0.9] degree, and a very high diagnostic value when it was above 0.9. Meta-

regression analysis and subgroup analysis were then used to explore the sources of heterogeneity between the included studies. In the literature of the included studies, most of the data sets were divided into training and test sets to build models of AI-assisted diagnostic systems for lung cancer diagnosis. The training set is generally used to train the AI model's ability to diagnose lung cancer, and the test set is used to test the AI model's ability to diagnose lung cancer. In this study, the power of various AI models to diagnose lung cancer was studied, and therefore only the data from the test set was referred to for analysis. If the samples of part of the study did not distinguish between the training set and the test set, the entire selection was used for research by default.

## 3 Results

### 3.1 Literature search, screening process, and results

In total 3156 articles were obtained according to the search strategy from databases China Biomedical Literature Database (CBM), Wanfang Database, China Knowledge Network (CNKI), PubMed, EMbase, and the Cochrane Library, and 0 articles were obtained through other databases. After de-duplication, 2811 reports were obtained. From the initial screening of 469 articles according to the inclusion and exclusion criteria, 14 papers were accepted for inclusion in the qualitative analysis according to the inclusion and exclusion criteria. The literature included 5251 patients, including 2 papers in Chinese and 12 papers in English (Fig 2).

### 3.2 Basic characteristics of the included studies and results of the quality assessment

The information of 28 included studies were concluded in Tables 2 and 3. To evaluate the quality of the 14 included studies, the risk of bias and the degree of relevance was assessed with the help of the QUADAS-2 tool combined with Revman software. In terms of index testing, about 40% of studies had a high risk of bias, and about 25% of the literature had an unclear risk of bias; in terms of the reference standard, about 40% of studies had an unclear risk of bias; and in terms of time, about 30% of studies had an unclear risk of bias (Fig 3).

### 3.3 Meta-analysis results

To select an appropriate effect model for meta-analysis, the level of heterogeneity was measured by calculating the $I^2$ value. If $I^2 > 50\%$, a high level of heterogeneity was considered to exist among the included literature and a random-effects model should be used for meta-analysis, if $I^2 < 50\%$, a low level of heterogeneity was thought to live among the included literature meta-analysis could be performed using a fixed-effects model. The results showed that $I^2 = 86.9\%$ and $I^2 > 50\%$, indicating a high heterogeneity among the included studies, and a random-effects model should be used for meta-analysis.

To investigate the diagnostic value of AI-assisted diagnosis for lung cancer diagnosis, the AI-assisted diagnosis system was evaluated by its merger sensitivity, merger specificity, positive likelihood ratio, negative likelihood ratio, diagnostic ratio, and AUC. The sensitivity and specificity of AI-assisted diagnosis were 0.87 [95% CI (0.82, 0.90)] and 0.87 [95% CI (0.82, 0.91)] respectively (Table 4, Figs 4 and 5), the underdiagnosis and misdiagnosis rates were 13% and 13% respectively. The AI-assisted diagnostic system had a high identification rate for lung cancer patients as well as non-lung cancer patients, and the diagnostic value of diagnosing lung cancer was high. According to the combined effect size of the AI-assisted diagnostic system for diagnosing lung cancer, the positive likelihood ratio and negative likelihood ratio were 6.5 [95% CI (4.6, 9.3)] and 0.15 [95% CI (0.11, 0.21)], respectively (Table 4), indicating that the

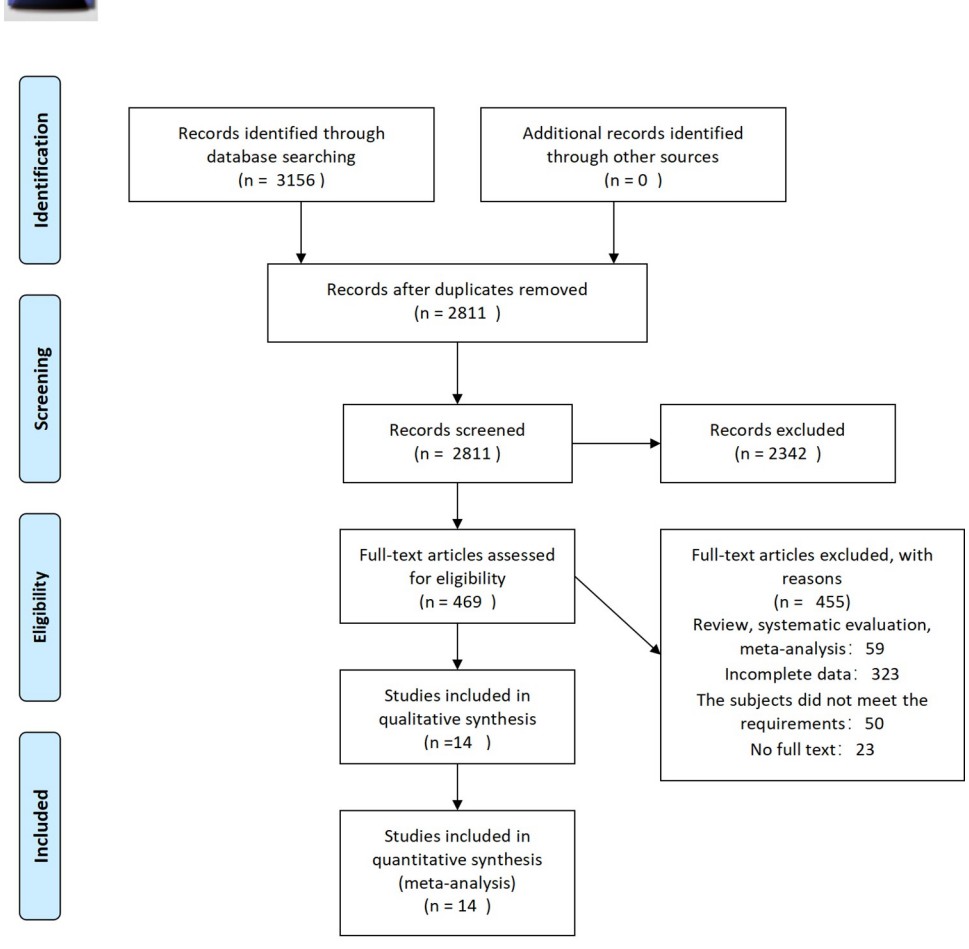

**Fig 2. Literature selection process.**

AI-assisted diagnostic system had a high diagnostic value. For diagnosing lung cancer, patients were 6.5 times more likely to have a correct result than an incorrect result, and diagnosing non-lung cancer patients was 0.15 times more likely to have an incorrect result than a correct result, with a higher likelihood of AI being able to correctly diagnose lung cancer patients as well as non-lung cancer patients. Based on the combined effect size of the AI-assisted diagnostic system for lung cancer diagnosis, the diagnostic ratio of the AI-assisted diagnostic system for lung cancer diagnosis in this study was 43 [95% CI (24, 76)] (Table 4), the combined diagnostic ratio can indicate to some extent the strength of the relationship between diagnostic experimental results and disease, the higher the diagnostic ratio, the stronger the association, the combined DOR of this study was 43, indicating that AI-assisted diagnosis has a high diagnostic value for lung cancer. According to the SROC curve of AI-assisted diagnosis of lung

**Table 2. Basic characteristics of the included studies.**

| Inclusion in the study | Country | Source | Total number of patients with lung cancer | Extraction characteristics | Gold Standard |
|---|---|---|---|---|---|
| Chamberlin 2021 [26] | United States | Patients who received routine lung cancer screening between January 2018 and July 2019 | 117 | | Doctor's diagnosis |
| Sun 2013 [27] | China | 4 hospitals in 2009 | 228 | 488 textural features | Pathology |
| Teramoto 2019 [28] | Japan | Patients suspected of having lung cancer | 25 | 25 features | Pathology or follow up |
| Wang 2016 [29] | China | LIDC-IDRI database | 322 | 150 quantitative image features | Doctor's diagnosis |
| Yin-Chen Hsu 2020 [30] | China | Asymptomatic participants at Evergreen Memorial Hospital, Chiayi, Taiwan, between February 2017 and August 2018 | 836 | | Pathology |
| Li Tian 2020 [31] | China | Patients with pulmonary nodules diagnosed at the Third Affiliated Hospital of Jinzhou Medical University from July 2017-July 2019 | 120 | | Pathology |
| Xu Liping 2014 [32] | China | March 2005-July 2006 Patients of the First Affiliated Hospital of Zhengzhou University | 59 | 21 radiological features+5 clinical parameters | Pathology |
| Dilger 2015 [33] | United States | The NLST and the Chronic Obstructive Pulmonary Disease Genetic Epidemiology (COPDGene) study | 50 | Intensity, shape, border, and texture | Pathology or follow up |
| Ren 2019 [34] | China | Lung Image Database Consortium (LIDC) of Image Database Resource Initiative (IDRI) | 1018 | Nodule intensity, texture, and morphology | Doctor's diagnosis |
| Duan 2020 [35] | China | First Affiliated Hospital of Zhengzhou University | 842 | Three grayscale features, seven morphological features and five texture features | Doctor's diagnosis |
| Li 2018 [36] | China | LIDC dataset and General Hospital of Guangzhou Military Command (GHGMC) dataset | 1318 | Intensity, texture, and geometric feature | Pathology |
| Manikandan2016 [37] | India | Patients of Bharat Scan center | 106 | 2-D shape feature, 3-D Centroid and Texture features | Doctor's diagnosis |
| Silva 2017 [38] | Brazil | LIDC-IDRI database | 200 | | Doctor's diagnosis |
| Dilger 2013 [39] | United States | University of Iowa Hospital and Medical Center+21 clinics in the United States research center | 10 | 36 texture and substantive features | Pathology |

cancer, the AUC was 0.93[95% CI (0.91, 0.95)] (Fig 6), indicating the high diagnostic value of AI for lung cancer.

To explore the sources of heterogeneity among the included literature, meta regressions were conducted the sample size of the test group was greater than 50 as covariates. The results of the regression analysis showed that the p-value of the group with a sample size greater than 50 was >0.5 (Table 5), indicating that the sample size greater than 50 was not associated with heterogeneity among the included literature.

To investigate the relationship between the types of AI algorithms and the heterogeneity among the included literature, subgroup analysis was performed according to the kinds of AI algorithms (support vector machine, artificial neural network, random forest, and other algorithms), and the combined diagnostic ratio was used as the effect size to explore the source of heterogeneity. The results showed that the combined diagnostic ratio was significantly lower in the support vector machine group than in other groups (Fig 7), i.e., the use of support vector machine algorithms to assist in the diagnosis of benign malignancy in lung tumors was less effective, and studies using support vector machine algorithms were a possible source of heterogeneity.

To test whether there was a publication bias in this study, a DEEK funnel plot was created by Stata 16.0 software. The results showed P = 0.33 (P>0.05) (Fig 8), indicating that there was

**Table 3. Diagnostic features.**

| Inclusion in the study | Year of publication | AI algorithms | Total sample size | TP | FP | FN | TN |
|---|---|---|---|---|---|---|---|
| Sun | 2013 | Support vector machines | 33 | 15 | 2 | 2 | 14 |
| Teramoto | 2019 | Random Forest | 43 | 24 | 13 | 1 | 5 |
| Wang | 2016 | Support vector machines | 193 | 91 | 15 | 31 | 56 |
| Yin-Chen Hsu | 2020 | Artificial Neural Network ANN | 234 | 6 | 34 | 2 | 192 |
| Li Tian | 2020 | Computer-aided diagnosis | 109 | 65 | 10 | 30 | 4 |
| Xu Liping | 2014 | fuzzy neural network | 44 | 19 | 2 | 2 | 21 |
| Dilger | 2015 | Artificial neural network | 50 | 20 | 2 | 2 | 26 |
| Dilger | 2015 | Linear discriminant analysis | 50 | 17 | 3 | 5 | 25 |
| Manikandan | 2016 | Support vector machine | 257 | 22 | 16 | 0 | 219 |
| Silva | 2017 | Convolutional neural network | 200 | 98 | 9 | 2 | 91 |
| Li | 2018 | Random forest1 | 100 | 17 | 13 | 3 | 63 |
| Li | 2018 | Random forest2 | 200 | 62 | 22 | 8 | 108 |
| Li | 2018 | Random forest3 | 300 | 52 | 22 | 6 | 220 |
| Li | 2018 | Random forest4 | 400 | 120 | 16 | 16 | 248 |
| Li | 2018 | Random forest5 | 500 | 147 | 31 | 13 | 309 |
| Li | 2018 | Random forest6 | 600 | 184 | 40 | 16 | 360 |
| Ren | 2019 | Manifold regularized classification deep neural network | 245 | 70 | 8 | 16 | 151 |
| Ren | 2019 | Classification deep neural network | 245 | 54 | 10 | 32 | 149 |
| Dilger | 2013 | Artificial neural network | 27 | 10 | 2 | 0 | 15 |
| Duan | 2020 | Artificial neural network1 | 204 | 84 | 14 | 44 | 62 |
| Duan | 2020 | Support vector machine1 | 204 | 88 | 31 | 40 | 45 |
| Duan | 2020 | Artificial neural network2 | 78 | 43 | 5 | 3 | 27 |
| Duan | 2020 | Support vector machine2 | 78 | 40 | 7 | 6 | 25 |
| Duan | 2020 | Artificial neural network3 | 33 | 15 | 2 | 1 | 15 |
| Duan | 2020 | Support vector machine3 | 33 | 14 | 3 | 2 | 14 |
| Chamberlin | 2021 | Artificial neural network ANN | 117 | 69 | 0 | 14 | 34 |

TP = true positive. FP = false positive. TN = true negative. FN = false negative.

no publication bias. The above results suggest that the AI-aided diagnosis system has a high diagnostic value for lung cancer and can be used to diagnose lung cancer in clinical practice.

## 4 Discussion

Lung cancer is a malignant tumor originating from the mucosa or glands of the bronchi and has a high prevalence and mortality rate worldwide. Currently, CT of the chest is the most

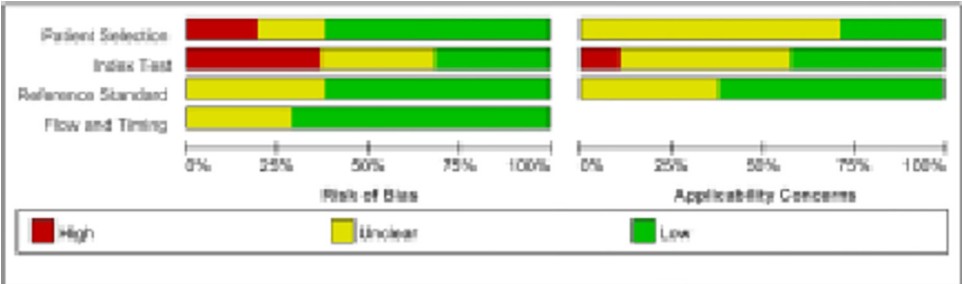

**Fig 3. Results of the quality evaluation of the included literature.** Red represents high degree of bias, yellow represents unclear and green represents a low degree of preference.

**Table 4. Combined effect sizes for AI-assisted diagnostic systems for the diagnosis of lung cancer.**

| Parameter | Estimate | 95% CI |
|---|---|---|
| Sensitivity | 0.87 | [0.82, 0.90] |
| Specificity | 0.87 | [0.82, 0.91] |
| Positive Likelihood Ratio | 6.5 | [4.6, 9.3] |
| Negative Likelihood Ratio | 0.15 | [0.11, 0.21] |
| Diagnostic Odds Ratio | 43 | [24, 76] |

commonly used tool for lung cancer screening. Its high resolution can show the relationship of adjacent organs and blood vessels more clearly and plays a unique advantage in the early screening of lung cancer [40]. However, specific benign lesions, such as inflammation, tuberculosis, and necrosis, and some textures in the lung images, as well as some objective factors such as the experience of the film reviewer, may affect the accuracy of this method and may

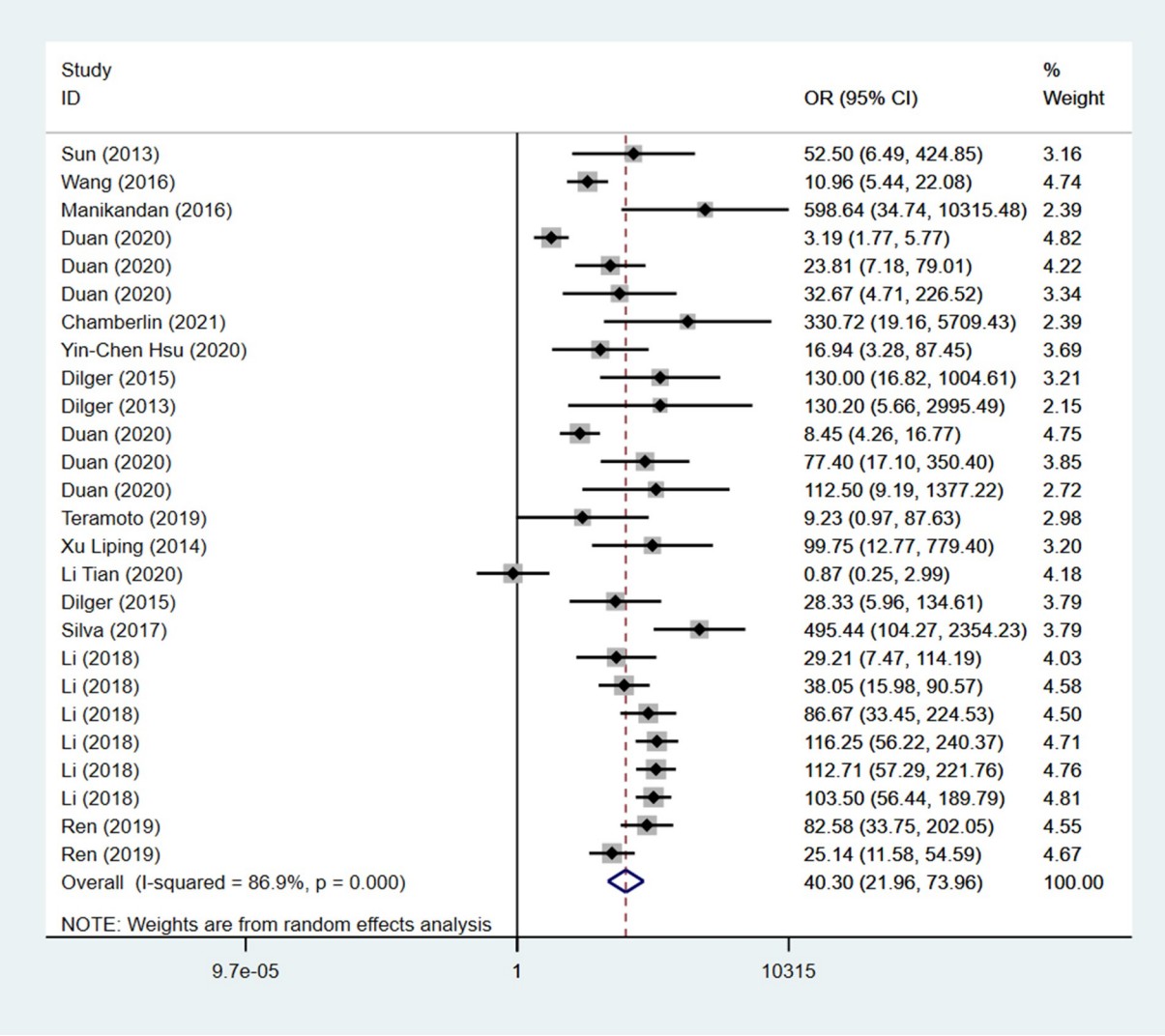

**Fig 4. OR forest plot of AI-assisted diagnostic system for lung cancer diagnosis.**

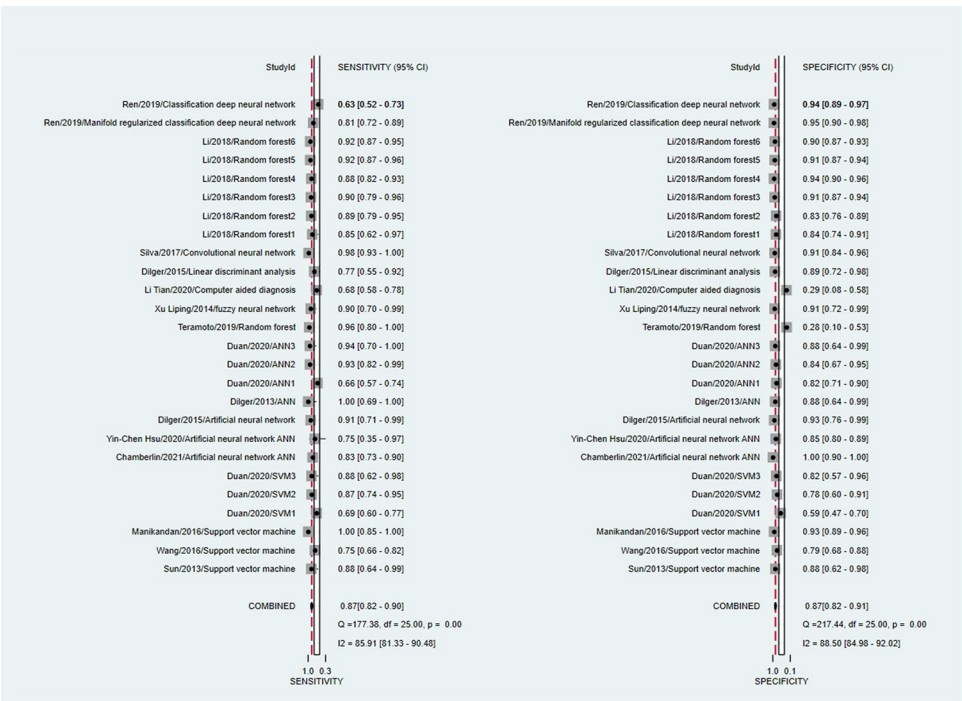

**Fig 5. Forest plot of sensitivity and specificity of AI-aided diagnosis for lung cancer diagnosis.**

easily lead to misdiagnosis and omission. Since applying AI-assisted diagnostic systems to clinical work, a new page has been turned in the study of a lung cancer diagnosis. Some studies have shown [41–43] that AI technologies are being used more and more extensively in the practice of clinical diagnosis and treatment. These technologies are mainly applied to diagnose and analyze various medical images such as skin lesions, pathological microscopic images, and radiological data, and AI technologies are remarkable in their ability to improve diagnostic accuracy, stability, and work efficiency. The use of AI-assisted diagnosis of lung cancer has now become commonplace in clinical research and work.

The results of the meta-analysis showed that the combined sensitivity of the AI-aided diagnosis system for lung cancer diagnosis was 0.87 [95% CI (0.82, 0.90)], specificity was 0.87 [95% CI (0.82, 0.91)] (CI stands for confidence interval.), the missed diagnosis rate was 13%, the misdiagnosis rate was 13%, the positive likelihood ratio was 6.5 [95% CI (4.6, 9.3)], the negative likelihood ratio was 0.15 [95% CI (0.11, 0.21)], a diagnostic ratio of 43 [95% CI (24, 76)] and a sum of area under the combined subject operating characteristic (SROC) curve of 0.93 [95% CI (0.91, 0.95)]. Based on the results, the AI-assisted diagnostic system for CT (Computerized Tomography), imaging has considerable diagnostic accuracy for lung cancer diagnosis, which is of significant value for lung cancer diagnosis and has greater feasibility of realizing the extension application in the field of clinical diagnosis. In addition, the results of this study are also consistent with previous literature [44–46] reports, indicating that the results of this study have reference value, AI with the help of a deep learning model, can compensate for the missed diagnosis caused by physicians' inexperience or incompetence, reduce the false-positive rate, and can improve the work efficiency to a certain extent. Therefore, this study suggests increasing the promotion of AI application in the clinical diagnosis of lung cancer.

AI-assisted diagnosis using different algorithms has different diagnostic outcomes. It has been shown [47] that traditional shallow learning algorithms are more advantageous for

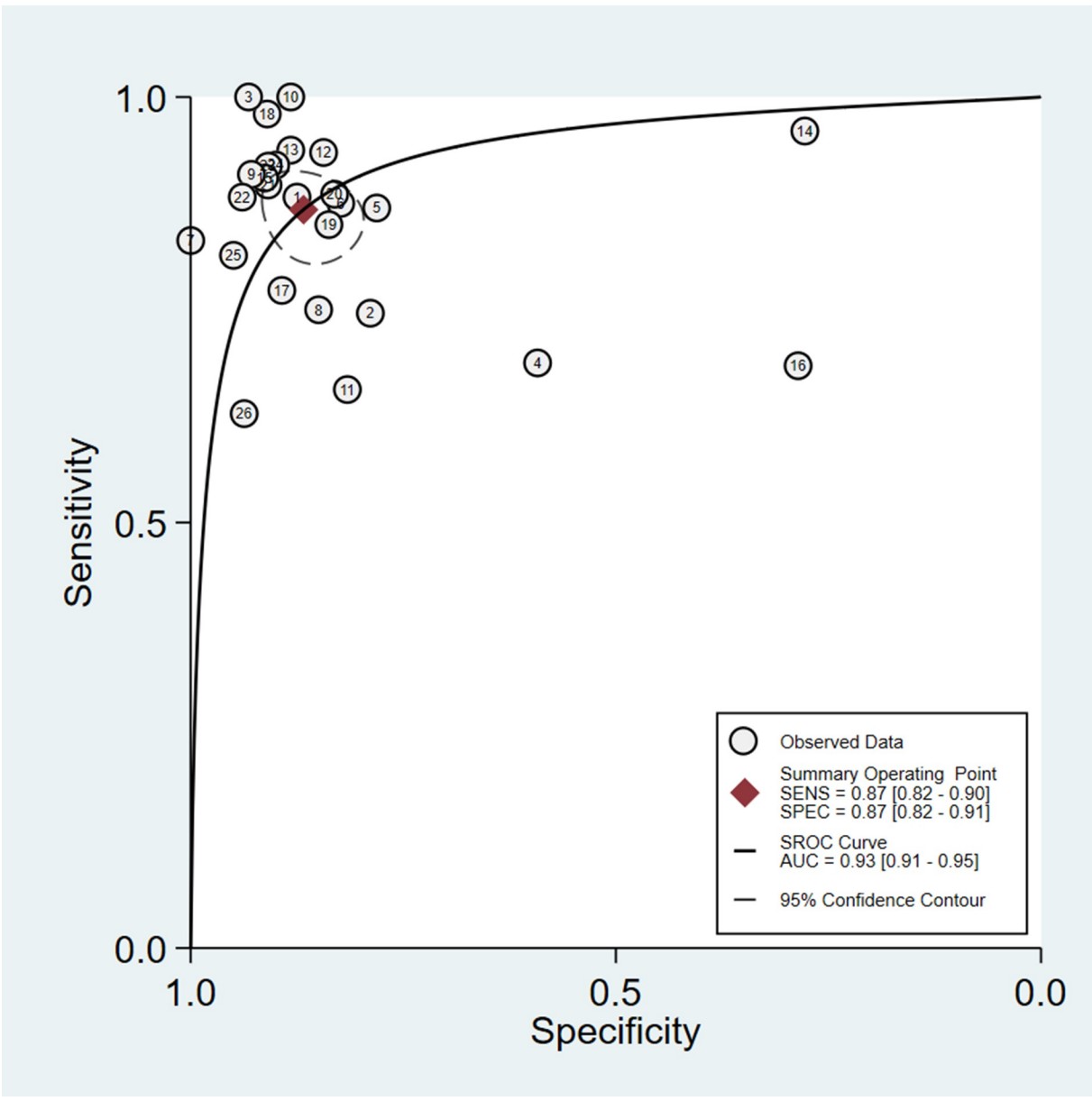

**Fig 6. SROC curve for AI-assisted diagnosis of lung cancer.**

minor sample diseases than deep learning algorithms. However, for lung cancer lung nodules, a condition with a large amount of sample data, deep learning algorithms improve the

**Table 5. Results of STATA regression analysis for AI-assisted diagnosis of lung cancer.**

| Logor | Coef. | Std. Err. | t | P>t | [95% Conf. Interval] |
|---|---|---|---|---|---|
| Total 50 | -2.398581 | 8.718715 | -0.28 | 0.786 | [-20.39312, 15.59596] |
| _cons | 2.403933 | 8.679412 | 0.28 | 0.784 | [-15.50949, 20.31936] |

Total 50: The group with a sample size greater than 50; Logor: logarithm of Odds ratio; Coef.: Estimation coefficient; Std. Err.: Standard error; t: T test value; P: p value; 95% Conf.Interval: 95% confidence interval; _cons: constant.

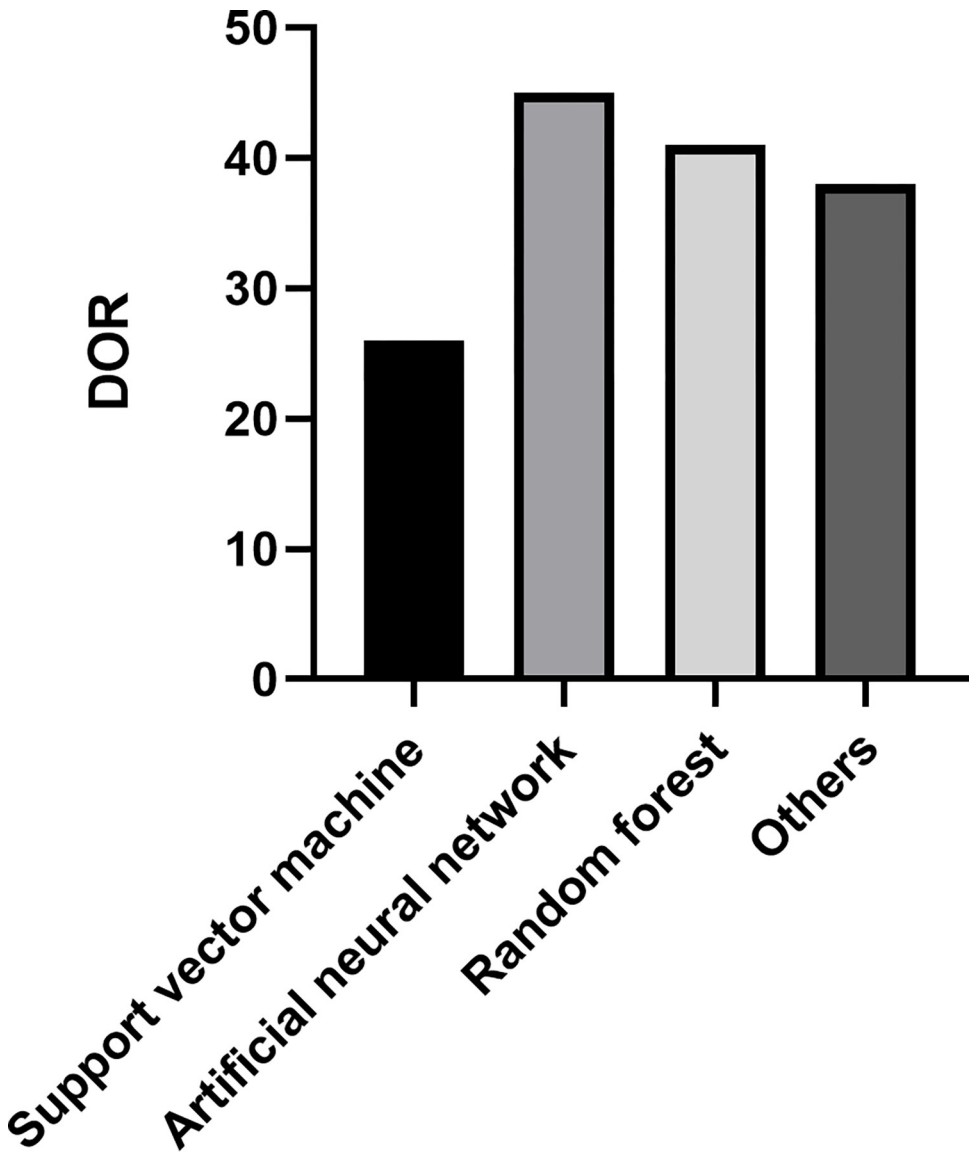

**Fig 7. Results of the subgroup analysis of lung cancer diagnosed with the aid of artificial lung cancer diagnosis.**

diagnostic accuracy of lung cancer more than shallow learning algorithms improve the diagnostic accuracy of lung cancer. Different algorithms have different diagnostic capabilities, especially radionics and deep learning, which can help identify not only the benignity and malignancy of lung nodules but even predict the aggressiveness and the prognosis of small cell lung cancer of the lung. The results of the meta-regression analysis in this study showed that whether the sample size was greater than 50 were not possible sources of heterogeneity. The subgroup analysis concluded that the diagnostic value of support vector machine for lung cancer was significantly lower than that of other algorithms since the literature on AI-assisted diagnostic studies of lung cancer using support vector machine is small, so the results of this subgroup analysis may be related to study on support vector machine algorithm.

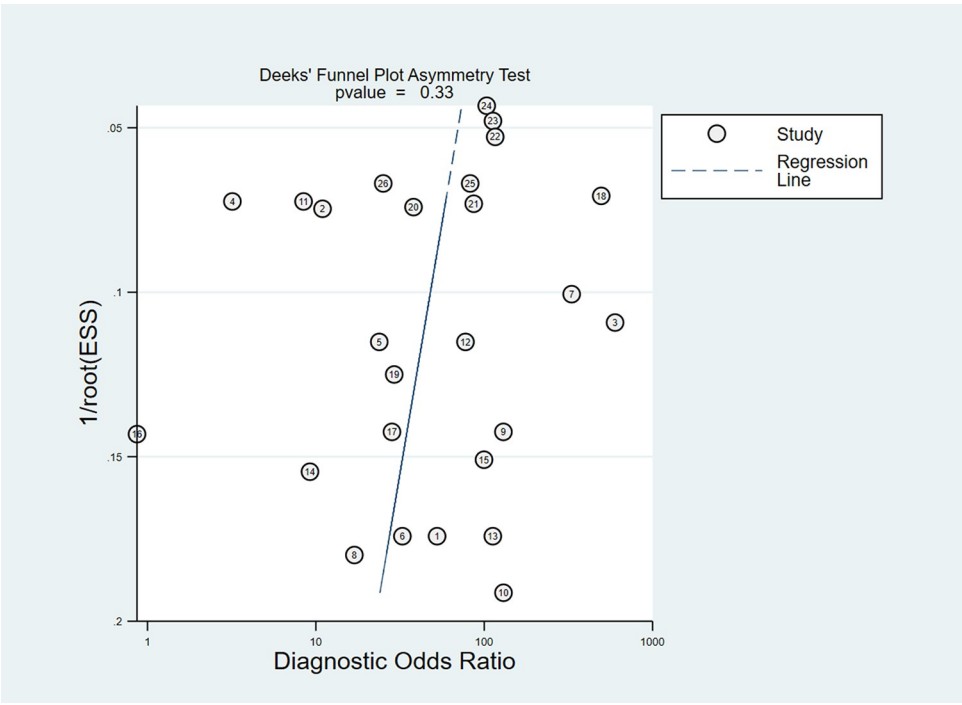

**Fig 8. Publication bias in AI-assisted diagnosis of lung cancer.**

This study also has certain limitations: (1) The high heterogeneity among the original studies included in this study may also be related to the different sources of study subjects, the large sample size gap between studies, and the varying number of features extracted by AI, and the results are subject to further study. (2) This study excluded literature for which diagnostic data were not available in full, which may have biased the results. (3) A comprehensive search of relevant databases was conducted in this study, but only Chinese and English literature were included, which may be subject to language bias. (4) The original studies included in this study were mainly retrospective, and the quality of the original studies will affect the quality of the systematic evaluation.

Although the effectiveness of AI in lung cancer diagnosis has been initially verified, most of the advances in AI pathology diagnosis at this stage are still at the laboratory research stage and have not entered the clinic. Its limitations are manifested in image data quality, data integration, complex pathology diagnosis, legal liability definition, and cost of use. With the advancement of AI and digital pathology technology, more and more experienced pathologists are involved in AI for lung cancer pathology image annotation. It is believed that AI diagnostic systems will play a more significant role in the accurate pathology diagnosis of lung cancer.

In summary, the results of this study show that the AI-aided diagnosis system based on CT images has a high value in the diagnosis of lung cancer and can be promoted as a method to diagnose lung cancer in clinical applications. Integrating various data such as CT images, pathology, patient's history, clinical features, physician's diagnosis, and patient follow-up into the AI-assisted diagnosis system for all-round evaluation of patients is the future direction of AI development, which will not only improve the diagnostic accuracy of lung cancer and reduce physician's workload but may also change the current medical model and promote the balanced development of medical resources in China.

## Supporting information

**S1 Checklist. PRISMA 2009 checklist.**
(DOC)

**S1 Data. Minimum data set.**
(DOCX)

**S1 Appendix.**
(ZIP)

## Author Contributions

**Data curation:** Mingsi Liu, Jinlin Guo.

**Investigation:** Jinghui Wu.

**Methodology:** Lin Zhang.

**Resources:** Jinghui Wu, Shulin Liu.

**Software:** Mingsi Liu, Yujiao Bai.

**Supervision:** Ke Tao.

**Writing – original draft:** Mingsi Liu, Jinghui Wu.

**Writing – review & editing:** Jinghui Wu, Nian Wang, Xianqin Zhang.

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
