## [Decision Letter · Decision Letter 0]

14 Oct 2022

PONE-D-22-21806The value of artificial intelligence in the diagnosis of lung cancer: a systematic review and meta-analysisPLOS ONE

Dear Dr. ke,

Thank you for submitting your manuscript to PLOS ONE. After careful consideration, we feel that it has merit but does not fully meet PLOS ONE’s publication criteria as it currently stands. Therefore, we invite you to submit a revised version of the manuscript that addresses the points raised during the review process.

Based on reviewer comments it is recommended to revisit the Introduction and Materials and Methods section to make significant changes. There are also major issues with references that needs to be addressed. 

We look forward to receiving your revised manuscript.

Kind regards,

Rahul Gomes, Ph.D.

Academic Editor

PLOS ONE

Journal Requirements:

 “This study was supported by two national natural science foundation of China projects (grant no. 32170119,31870135).”

4. We note you have included a table to which you do not refer in the text of your manuscript. Please ensure that you refer to Table 2 in your text; if accepted, production will need this reference to link the reader to the Table.

Reviewers' comments:

Reviewer's Responses to Questions

**Comments to the Author**

1. Is the manuscript technically sound, and do the data support the conclusions?

Reviewer #1: Yes

Reviewer #2: Yes

Reviewer #3: Partly

2. Has the statistical analysis been performed appropriately and rigorously? 

Reviewer #1: Yes

Reviewer #2: N/A

Reviewer #3: Yes

3. Have the authors made all data underlying the findings in their manuscript fully available?

Reviewer #1: Yes

Reviewer #2: Yes

Reviewer #3: No

4. Is the manuscript presented in an intelligible fashion and written in standard English?

Reviewer #1: Yes

Reviewer #2: Yes

Reviewer #3: Yes

5. Review Comments to the Author

Reviewer #1: No comment

Reviewer #2: Comments and Suggestions for Authors

# Major comments

I list my major comments on the manuscript below.

1- The innovation and the motivation behind this work are not clearly highlighted. Please work on this and prove to us why this work is valuable. The novelty of the proposed model and the contributions of this paper are questionable. The proposal is poorly defined.

2- The introduction section should follow the state of the art of this field and review what has been done, for supporting the research gap and the significance of this study. Please improve the state-of-the-art overview, to clearly show the progress beyond the state of the art.

3- The contributions of this paper are not listed in the introduction section.

4- The paper's organization must be added at the end of the introduction section.

5- Capitalize the first letters of the words of the abbreviations.

6- The authors used some abbreviations without clarifying the original words of these abbreviations, for example, but not limited to CT, CI, etc.

7- Some of the symbols are not clear to the reader what they stand for. Therefore, a table of all symbols used in the paper will enhance the readability of the paper.

8- 1.4 Study content must be 1.3 Study content.

9- There are some claims that need to be supported by references. Review it.

10- The used Platform configurations for testing the proposed model are not stated.

11- The evaluation metrics must be presented in a separate section before usage. In this section, the types of the used metrics are provided. In addition, the mechanism of evaluation (i.e., The meaning of the higher and lower values of the used metrics).

12- There are recent references not included, there are only three references (2021) and no references (2022)

Reviewer #3: In general, the review is well written. However, I have some major concerns to share:

Based on the final results of the query, authors ended up with 21 papers, 16 are written in Chinese and 5 in English. In the world of publication/research, English is considered the ‘official’ language - for many reasons. Some interested readers wouldn’t be able to refer to those 16 papers for further reading. Consequently, one of the critical criteria of the query in such a manuscript should be the language of the manuscript to be set to English.

In tables 1 and 1 (or at least in one of them), it is important to include the number of the corresponding reference. Speaking of which, are the 21 papers cited in the references section? I am pretty sure they are not!!!!! (This is weird)

In the first two big paragraphs of the “discussion” do not appear to me as ‘discussion’. They should be somewhere in the “introduction”, or “Literature review”.

Some minor comments:

- In the Introduction section, first paragraph, row 4 “Statistics show that stage 0 patients diagnosed…”, I think is missing here. In the same line, authors used a 25-years old paper to mention a statistics concerning the survival rate!

- In the Introduction section, the second paragraph, rows 4 and 5, acronyms should be explained in their first instances.

- In the Introduction section, the third paragraph, before the last row, a period to be omitted.

- On page 2, section 1.2 “AI has made breakthroughs in detecting, diagnosing,

- and treating lung cancer”. I am not sure about “treating”.

- On page 2, section 1.2, it is either DCNN or CNN not DCN!

- Section 2.5, is “[8]” a citation? If yes, it should be consistent, i.e., using superscript.

- The references are not written in a consistent manner (e.g., some authors’ names are written in capital letters…).

6. PLOS authors have the option to publish the peer review history of their article (what does this mean?). If published, this will include your full peer review and any attached files.

Reviewer #1: **Yes: **Atheer Alrammahi

Reviewer #2: **Yes: **HOSAM ALRAHHAL

Reviewer #3: No

---

## [Author Response · Author response to Decision Letter 0]

29 Nov 2022

Dear reviewer:

Thank you for your decision and constructive comments on my manuscript. We have carefully considered the suggestion of Reviewer and make some changes. We have tried our best to improve and made some changes in the manuscript.

The blue part that has been revised according to your comments. Revision notes, point-to-point, are given as follows:

Journal Requirements:

1.Please ensure that your manuscript meets PLOS ONE's style requirements, including those for file naming.

I'm sorry that we ignored the style requirements of the magazine when writing the manuscript. Now we have revised it according to the style template provided by the magazine.

2. Thank you for stating the following financial disclosure: “This study was supported by two national natural science foundation of China projects (grant no. 32170119,31870135).”Please state what role the funders took in the study. If the funders had no role, please state: "The funders had no role in study design, data collection and analysis, decision to publish, or preparation of the manuscript.". If this statement is not correct you must amend it as needed. Please include this amended Role of Funder statement in your cover letter; we will change the online submission form on your behalf.

The role of the funder in the study has been added to the online submission form, please check.

Upon re-submitting your revised manuscript, please upload your study’s minimal underlying data set as either Supporting Information files or to a stable, public repository and include the relevant URLs, DOIs, or accession numbers within your revised cover letter.

It was an oversight on our part not to account for the minimal data set of results described in the manuscript, which we have now uploaded as a supporting information file.

4. We note you have included a table to which you do not refer in the text of your manuscript. Please ensure that you refer to Table 2 in your text; if accepted, production will need this reference to link the reader to the Table.

Due to our oversight, Table 2 included in the manuscript is indeed not described in the text, and is now added in the first line of Part 3.2.

Response to Reviewer #2

1-The innovation and the motivation behind this work are not clearly highlighted. Please work on this and prove to us why this work is valuable. The novelty of the proposed model and the contributions of this paper are questionable. The proposal is poorly defined.

As for the innovation and motivation behind the work, the manuscript does describe less, which has been added in part 1.4.

2- The introduction section should follow the state of the art of this field and review what has been done, for supporting the research gap and the significance of this study. Please improve the state-of-the-art overview, to clearly show the progress beyond the state of the art.

In the original manuscript, the latest development in this field and the completed work are put in the discussion section. After considering the comments of reviewers, we think that the structural design of the paper is not reasonable, and now we have transferred this part to 1.3 Research Progress.

3- The contributions of this paper are not listed in the introduction section.

Contributions to this paper are listed in section 1.4 of introduction.

4- The paper's organization must be added at the end of the introduction section.

The paper's organization has been added at the end of the introduction. We understand the paper's organization as an outline. If you want to express the meaning of the directory or technical route, we can change it again.

5- Capitalize the first letters of the words of the abbreviations.

The initials of the acronym have been capitalized.

6- The authors used some abbreviations without clarifying the original words of these abbreviations, for example, but not limited to CT, CI, etc.

The abbreviations of the original words that we omitted have been supplemented.

7- Some of the symbols are not clear to the reader what they stand for. Therefore, a table of all symbols used in the paper will enhance the readability of the paper.

Since the table in the paper is directly output by the analysis software, it is not explained. Now it has been added in Table 4 and Table 5.

8- 1.4 Study content must be 1.3 Study content.

After the reviewer's reminding, we have found the error caused by our negligence, and have reordered all parts of the article.

9- There are some claims that need to be supported by references. Review it.

We examine the article from beginning to end, re-cite the relevant literature as needed to support the claims.

10- The used Platform configurations for testing the proposed model are not stated.

The platform configurations used to test the proposed model in this paper are labeled in sections 2, 2.3, 2.4, and 2.5, respectively.

11- The evaluation metrics must be presented in a separate section before usage. In this section, the types of the used metrics are provided. In addition, the mechanism of evaluation (i.e., The meaning of the higher and lower values of the used metrics).

The evaluation indicators and mechanism of this study have been presented separately in Part 2.5.

12- There are recent references not included, there are only three references (2021) and no references (2022)

After examination, the reference to the latest references was indeed ignored. Now, references 1, 2, 9, 14, 17, 22, 24, 37, 41, 47 have been added.

Response to Reviewer #3

1-Based on the final results of the query, authors ended up with 21 papers, 16 are written in Chinese and 5 in English. In the world of publication/research, English is considered the ‘official’ language - for many reasons. Some interested readers wouldn’t be able to refer to those 16 papers for further reading. Consequently, one of the critical criteria of the query in such a manuscript should be the language of the manuscript to be set to English.

It can be seen that both the reviewers and us believe that the paper needs to cite international papers, namely papers from other countries, so that the paper can be placed in an international research background. The Chinese literature selected by the research is indeed needed, and these articles have English titles and English abstracts for interested readers to consult.

2-In tables 1 and 1 (or at least in one of them), it is important to include the number of the corresponding reference. Speaking of which, are the 21 papers cited in the references section? I am pretty sure they are not!!!!! (This is weird)

Due to our negligence, the 21 papers that we did select were not cited and have been completed now.Such a mistake should not have happened. We will be more careful in the future.

3-In the first two big paragraphs of the “discussion” do not appear to me as ‘discussion’. They should be somewhere in the “introduction”, or “Literature review”.

After consideration, it is true that the definition of the discussion and introduction is vague. Now, the first two paragraphs of the discussion have been put into the introduction to describe the research progress.

4-In the Introduction section, first paragraph, row 4 “Statistics show that stage 0 patients diagnosed…”, I think is missing here. In the same line, authors used a 25-years old paper to mention a statistics concerning the survival rate!

Thanks to the reviewers for checking the details of the article, which put us to shame. At present, the errors found by reviewers have been corrected and the full text has been examined more closely. We supplemented and changed the articles on relevant statistical data, and replaced recent literatures to support the statistical data.

5-In the Introduction section, the second paragraph, rows 4 and 5, acronyms should be explained in their first instances.

We supplemented the explanation of the abbreviations in the article.

6-In the Introduction section, the third paragraph, before the last row, a period to be omitted.

We removed the period in the last line of the third paragraph.

7-On page 2, section 1.2 “AI has made breakthroughs in detecting, diagnosing, and treating lung cancer”. I am not sure about “treating”.

Views on the relationship between artificial intelligence and lung cancer treatment, cited in the references.

8-On page 2, section 1.2, it is either DCNN or CNN not DCN!

Such a typo is not correct. We have changed "DCN" to "DCNN".

9-Section 2.5, is “[8]” a citation? If yes, it should be consistent, i.e., using superscript.

[8] does refer to me and we have set it to superscript.

10-The references are not written in a consistent manner (e.g., some authors’ names are written in capital letters…).

We have unified the format of the references.

We would like to thank the editors and all the reviewing members for their valuable feedback. Looking forward to your reply.

Sincerely,

Ke Tao

College of Life Science , Sichuan University, Chengdu610041, China.

Email: taoke@scu.edu.cn

---

## [Decision Letter · Decision Letter 1]

19 Dec 2022

PONE-D-22-21806R1The value of artificial intelligence in the diagnosis of lung cancer: a systematic review and meta-analysisPLOS ONE

Dear Dr. ke,

Thank you for submitting your manuscript to PLOS ONE. After careful consideration, we feel that it has merit but does not fully meet PLOS ONE’s publication criteria as it currently stands. Therefore, we invite you to submit a revised version of the manuscript that addresses the points raised during the review process.

Please note that one of the reviewers has raised a concern about the breadth of this research in terms of where it originated from. The reviewer mentioned that having only five papers originating globally in the span of 10 years raises concern about the extent of this review article. I would encourage you to provide a better explanation for this issue. You can also modify your review decision criteria to make readers aware about this limitation. 

We look forward to receiving your revised manuscript.

Kind regards,

Rahul Gomes, Ph.D.

Academic Editor

PLOS ONE

Journal Requirements:

Reviewers' comments:

Reviewer's Responses to Questions

**Comments to the Author**

1. If the authors have adequately addressed your comments raised in a previous round of review and you feel that this manuscript is now acceptable for publication, you may indicate that here to bypass the “Comments to the Author” section, enter your conflict of interest statement in the “Confidential to Editor” section, and submit your "Accept" recommendation.

Reviewer #1: All comments have been addressed

Reviewer #2: (No Response)

Reviewer #3: (No Response)

2. Is the manuscript technically sound, and do the data support the conclusions?

Reviewer #1: Yes

Reviewer #2: Yes

Reviewer #3: (No Response)

3. Has the statistical analysis been performed appropriately and rigorously? 

Reviewer #1: Yes

Reviewer #2: Yes

Reviewer #3: N/A

4. Have the authors made all data underlying the findings in their manuscript fully available?

Reviewer #1: Yes

Reviewer #2: Yes

Reviewer #3: Yes

5. Is the manuscript presented in an intelligible fashion and written in standard English?

Reviewer #1: Yes

Reviewer #2: Yes

Reviewer #3: Yes

6. Review Comments to the Author

Reviewer #1: No comments no comments no comments.

Reviewer #2: 1- Some of the symbols are not clear to the reader what they stand for. Therefore, a table of all symbols used in the paper will enhance the readability of the paper. It would be best if you had a separate table for all symbols in the article.

2- Review the English to correct some grammar mistakes.

Reviewer #3: I would like to thank the authors for their answers.

The authors have responded appropriately to my all concerns except for the first one:

My Comment:

1-Based on the final results of the query, authors ended up with 21 papers, 16 are written in Chinese and 5 in English. In the world of publication/research, English is considered the ‘official’ language - for many reasons. Some interested readers wouldn’t be able to refer to those 16 papers for further reading. Consequently, one of the critical criteria of the query in such a manuscript should be the language of the manuscript to be set to English.

Authors’ answer:

It can be seen that both the reviewers and us believe that the paper needs to cite international papers, namely papers from other countries, so that the paper can be placed in an international research background. The Chinese literature selected by the research is indeed needed, and these articles have English titles and English abstracts for interested readers to consult.

I am afraid, I am still not convinced. I definitely didn’t mean to oppose ‘International research’ the way the authors explained. My concerns are already mentioned in my comment above. I understand that – in some ‘rare’ cases - a couple of papers written in a non-English language could be unavoidable to miss since they are considered ‘key’ papers, but I am still not convinced that in such an active research area, only 5 papers written in English have been found to meet the criteria from 2010 to 2021 (10 years!!!).

7. PLOS authors have the option to publish the peer review history of their article (what does this mean?). If published, this will include your full peer review and any attached files.

Reviewer #1: **Yes: **Atheer Alrammahi

Reviewer #2: **Yes: **HOSAM ALRAHHAL

Reviewer #3: No

---

## [Author Response · Author response to Decision Letter 1]

19 Jan 2023

Dear reviewer:

Thank you for your decision and constructive comments on my manuscript. We have carefully considered the suggestion of Reviewer and make some changes. We have tried our best to improve and made some changes in the manuscript.

The red part that has been revised according to your comments. Revision notes, point-to-point, are given as follows:

Journal Requirements:

1.Please review your reference list to ensure that it is complete and correct. If you have cited papers that have been retracted, please include the rationale for doing so in the manuscript text, or remove these references and replace them with relevant current references. Any changes to the reference list should be mentioned in the rebuttal letter that accompanies your revised manuscript. If you need to cite a retracted article, indicate the article’s retracted status in the References list and also include a citation and full reference for the retraction notice.

We have reviewed and made the necessary changes to the reference list as per your request. The reference list is now complete and correct. We have added or replaced new references to studies numbered 41, 43, 24, 22, 10, and 33 to 39 in the new manuscript, and have removed studies numbered 2,30, 32 to 34, 36 to 41, and 43 to 47 in the old manuscript.

Response to Reviewer #2

1-Some of the symbols are not clear to the reader what they stand for. Therefore, a table of all symbols used in the paper will enhance the readability of the paper. It would be best if you had a separate table for all symbols in the article.

Thank you for your valuable feedback. We have added Table 5 on page 25 in the new manuscript. This table is used to explain the symbols that appear in the text.

2- Review the English to correct some grammar mistakes.

Thank you for your suggestions. We have reviewed the manuscript and corrected any grammatical errors.

Response to Reviewer #3

1-Based on the final results of the query, authors ended up with 21 papers, 16 are written in Chinese and 5 in English. In the world of publication/research, English is considered the ‘official’ language - for many reasons. Some interested readers wouldn’t be able to refer to those 16 papers for further reading. Consequently, one of the critical criteria of the query in such a manuscript should be the language of the manuscript to be set to English.

Thank you for your suggestions. After careful consideration, we have kept a portion of the valuable Chinese literature from the original 21 references and incorporated new English references. As a result, a total of 14 references have been included in the research, with 12 being English references and 2 being Chinese references.

Sincerely,

Ke Tao

College of Life Science , Sichuan University, Chengdu610041, China.

Email: taoke@scu.edu.cn

---

## [Decision Letter · Decision Letter 2]

6 Feb 2023

The value of artificial intelligence in the diagnosis of lung cancer: a systematic review and meta-analysis

PONE-D-22-21806R2

Dear Dr. ke,

We’re pleased to inform you that your manuscript has been judged scientifically suitable for publication and will be formally accepted for publication once it meets all outstanding technical requirements.

Kind regards,

Rahul Gomes, Ph.D.

Academic Editor

PLOS ONE

Additional Editor Comments (optional):

Reviewers' comments:

Reviewer's Responses to Questions

**Comments to the Author**

1. If the authors have adequately addressed your comments raised in a previous round of review and you feel that this manuscript is now acceptable for publication, you may indicate that here to bypass the “Comments to the Author” section, enter your conflict of interest statement in the “Confidential to Editor” section, and submit your "Accept" recommendation.

Reviewer #1: All comments have been addressed

Reviewer #3: All comments have been addressed

2. Is the manuscript technically sound, and do the data support the conclusions?

Reviewer #1: Yes

Reviewer #3: Yes

3. Has the statistical analysis been performed appropriately and rigorously? 

Reviewer #1: Yes

Reviewer #3: Yes

4. Have the authors made all data underlying the findings in their manuscript fully available?

Reviewer #1: Yes

Reviewer #3: Yes

5. Is the manuscript presented in an intelligible fashion and written in standard English?

Reviewer #1: Yes

Reviewer #3: Yes

6. Review Comments to the Author

Reviewer #1: Not comment Not comment Not comment

Reviewer #3: Thank you for the corrections.

The manuscript now looks more "universally" academic and scientific.

I would suggest - NOT obligatory - to write a couple of sentences stating that in the manuscript, AI means Machine learning (ML) and/or deep learning (DL), especially that some references already contain the term ML/DL.

7. PLOS authors have the option to publish the peer review history of their article (what does this mean?). If published, this will include your full peer review and any attached files.

Reviewer #1: **Yes: **Atheer Alrammahi

Reviewer #3: No

---

## [Editor Report · Acceptance letter]

10 Feb 2023

PONE-D-22-21806R2 

The value of artificial intelligence in the diagnosis of lung cancer: a systematic review and meta-analysis 

Dear Dr. Tao:

I'm pleased to inform you that your manuscript has been deemed suitable for publication in PLOS ONE. Congratulations! Your manuscript is now with our production department. 

Kind regards, 

on behalf of

Dr. Rahul Gomes 

Academic Editor

PLOS ONE